# Whole Blood Reactivity to Viral and Bacterial Pathogens after Non-Emergent Cardiac Surgery during the Acute and Convalescence Periods Demonstrates a Distinctive Profile of Cytokines Production Compared to the Preoperative Baseline in Cohort of 108 Patients, Suggesting Immunological Reprogramming during the 28 Days Traditionally Recognized as the Post-Surgical Recovery Period

**DOI:** 10.3390/biomedicines12010028

**Published:** 2023-12-21

**Authors:** Krzysztof Laudanski, Da Liu, Lioudmila Karnatovskaia, Sanghavi Devang, Amal Mathew, Wilson Y. Szeto

**Affiliations:** 1Department of Anesthesiology and Perioperative Care, Mayo Clinic, Rochester, MN 55905, USA; 2Department of Obstetrics and Gynecology, Shengjing Hospital of China Medical University, Shenyang 110055, China; liud1@sj-hospital.org; 3Division of Pulmonary and Critical Care, Department of Internal Medicine, Mayo Clinic, Rochester, MN 55905, USA; karnatovskaia.lioudmila@mayo.edu; 4Department of Critical Care Medicine, Mayo Clinic, Jacksonville, FL 32224, USA; sanghavi.devang@mayo.edu; 5School of Biomedical Engineering, Science and Health Systems, Drexel University, Philadelphia, PA 19104, USA; agm76@drexel.edu; 6Division of Cardiovascular Surgery, Department of Surgery, University of Pennsylvania, Philadelphia, PA 19104, USA; wilson.szeto@pennmedicine.upenn.edu

**Keywords:** cardiac surgery, immune system, inflammation, profiling, TNFα, M-CSF, IL-6, IL-10, IL-2, blood stimulation, reprogramming

## Abstract

The release of danger signals from tissues in response to trauma during cardiac surgery creates conditions to reprogram the immune system to subsequent challenges posed by pathogens in the postoperative period. To demonstrate this, we tested immunoreactivity before surgery as the baseline (t_baseline_), followed by subsequent challenges during the acute phase (t_24h_), convalescence (t_7d_), and long-term recovery (t_3m_). For 108 patients undergoing elective heart surgery, whole blood was stimulated with lipopolysaccharide (LPS), Influenza A virus subtype N2 (H3N2), or the Flublok™ vaccine to represent common pathogenic challenges. Leukocytosis, platelet count, and serum C-reactive protein (CRP) were used to measure non-specific inflammation. Cytokines were measured after 18 h of stimulation to reflect activation of the various cell types (activated neutrophils–IL-8; activated T cells-IL-2, IFNγ, activated monocyte (MO)-TNFα, IL-6, and deactivated or atypically activated MO and/or T cells–M-CSF, IL-10). IL-2 and IL-10 were increased at t_7d,_ while TNFα was suppressed at t_24h_ when LPS was utilized. Interestingly, M-CSF and IL-6 production was elevated at seven days in response to all stimuli compared to baseline. While some non-specific markers of inflammation (white cell count, IL-6, and IL-8) returned to presurgical levels at t_3m_, CRP and platelet counts remained elevated. We showed that surgical stimulus reprograms leukocyte response to LPS with only partial restoration of non-specific markers of inflammation.

## 1. Introduction

Mechanical destruction of tissues, exposure to artificial surfaces, and an environment rich in free radicals all challenge immunostasis during cardiac surgery [1,2,3,4,5,6]. These stimuli can also affect genetic mechanisms controlling immune system activation [7,8,9,10]. Consequently, surgery-triggered effects may persist well past the initial insult [5,11,12,13,14,15,16,17]. While sepsis trials demonstrate that early immune reprogramming determines its short- and long-term performance, similar data for cardiac surgery are lacking [17,18,19,20,21,22]. We previously demonstrated that post-surgical persistence of C-reactive protein (CRP) accompanied by an increase in cytomegalovirus (CMV) antibody titers suggests an acquisition of new immunostasis months after the initial surgery [23,24]. Interestingly, this may be mediated by a decreased ability of the whole blood to produce interferon γ (IFNγ) and interleukin 12 (IL-12) [25]. Persistence of these abnormalities in leukocyte responsiveness may account for the emergence of long-term morbidities in the aftermath of cardiac surgery [26,27,28,29,30,31,32,33,34].

Most cardiac surgery studies focus on short-term outcomes without documenting the immune system’s performance at baseline [2,8,9,10,13,35,36,37,38,39,40,41,42,43,44]. Serum biomarkers have been studied the most, yet functional performance of leukocytes may actually be more pertinent to recovery [1,2,7,8,10,13,14,23,24,27,29,37,38,39,40,43]. Most studies assessing leukocyte responsiveness utilized isolated cell populations, a cumbersome process subjected to methodological bias [7,9,10,24,29,45]. Isolated leukocyte stimulation in vitro lacks the inherent complexity of a biological system with a subsequent reduction in translational potential [46,47,48]. In contrast, utilizing whole blood stimulation delivers both research data and actionable clinical context [49,50,51,52]. Available in vitro studies of leukocyte responsiveness that involved whole blood used stimuli (ionomycin, phorbol 12-myristate 13-acetate, very high dose of LPS) that are difficult to translate into clinical context [46,52,53,54,55,56,57].

In this study, we therefore aimed to investigate short- and long-term immuno-reactivity after cardiac surgery using whole blood and translational-relevant stimuli. We hypothesized that the ability of the whole blood to produce cytokines would be enhanced shortly after cardiac surgery with the emergence of global suppression at seven days, followed by recovery to the presurgical baseline. We elected to measure pro-inflammatory (IL-6, TNFα, IL-2, IFNγ) and anti-inflammatory cytokines (IL-10, M-CSF) as their balance is critical to the recovery of immunostasis [8,18,30,35,36,55]. Cytokine production patterns were contrasted with the typical lab values of the immune system activation, including white blood count (WBC), platelet count, serum C-reactive protein, and interleukin-6 (IL-6) [5,13,36,38].

## 2. Materials and Methods

### 2.1. Consents

The University of Pennsylvania Institutional Review Board approved this prospective cohort study (#815686). All patients scheduled for non-emergent cardiac surgery were approached. Patients who provided consent were included in the study. 

### 2.2. Studied Population

We recruited a convenience sample of 108 patients undergoing cardiac surgery between April 2016 and March 2021. Adult patients (over 18 years old) scheduled for non-emergent cardiac surgery that employed cardiopulmonary extracorporeal bypass machine were included in the study. Exclusion criteria were lack of consent, emergent procedure, history of cancer in the last five years, and immunocompromised status, as documented in the electronic health record (EHR). 

### 2.3. Clinical Data Collection

Demographic and clinical data were obtained from EHR. These included surgical, anesthesia, and perioperative data. Morphine equivalents were calculated for opioids given in the first 24 h following surgery. The APACHE II score was used to gauge the severity of clinical status [58]. Diagnoses of cerebrovascular event, pulmonary embolism, and deep venous thrombosis pre- and peri-operatively were extracted from the EHR. Mortality was assessed at 28 days and 3 months. 

### 2.4. Study Procedure

After consent was obtained, the patient’s blood was collected prior to surgery (t_0_). Subsequent blood samplings took place 24 h (t_24h_) and 7 days (t_7d_) after the surgical procedure, with final follow-ups occurring at 3 months (t_3m_). Blood was collected from existing arterial and central venous lines during the hospital stay or was manually drawn using the Vacutainer™ system prefilled with heparin as anticoagulant (BD; Franklin Lakes, NJ, USA) and stored at 4 °C until further processing within 4 h of collection.

### 2.5. Stimulation of the Blood

A total of 500 µL of whole blood was stimulated with LPS [50 ng/mL] (Lonza, Wayne PA, USA), H3N2 [1 µg/mL] (BEI; Manassas, VA, USA), and FluBlok Quadrivalent vaccine (Protein Sciences Corporation, Meriden, CT, USA), or left without any addition for 18 h in the orbital shaker at 37 °C and 5% CO_2_ [59,60]. Following the stimulation period, the blood was spun for 5 min at 2000× *g* in a centrifuge, and plasma was collected. All cultures except LPS cultures were supplemented with polymixin B [50 IU/mL] (Millipore Sigma; St. Louis, MO, USA). The level of cytokines (IL-1β, IL-4, IL-5, IL-7, IL-8, IL-9, IL-10, IL-12p70, IL-13, IL-17a, IL-18, IL-27, IL-31, RANTES, MIP-1α, MCP-1, eotaxin, TNFα, TNFβ, IFNγ,) were analyzed using multiplex platform (Luminex, Northbrook, IL, USA) or enzyme-linked immunosorbent assay (ELISA) (SinoBiological; Wayne, NJ, USA)

### 2.6. Statistical Analysis

Parametric characteristics of the variables were assessed using Shapiro–Wilk W and K-S tests. Variables were reported as mean ± SD and compared using Student’s *t*-test or ANOVA, or as median (M_e_) and interquartile ranges (IR) with the U-Mann–Whitney test. For multiple variables, an ANOVA with two or more variables was used, with the *η*^2^ statistic determining significance of the association. When available, the data were analyzed as longitudinal samples with Duncan’s correction for multiple comparisons using two-way factor (time, stimulus) ANOVA. *p*-values less than 0.05 were considered statistically significant for both one- or two-tailed hypotheses. *r*^2^ correlation coefficients were used to assess relationships.

Considering the very high dynamic range of cytokine production, we prepared several figures with a Y logarithmic scale. Furthermore, in the figures, we used “*” (two-tailed hypothesis with *p* < 0.05 or), or “&” (one-tailed statistic if a specific null hypothesis was formulated) to compare samples longitudinally within their respective stimuli. “#” denotes the significance level below 0.05 in two-tailed testing while comparing cytokine production to unstimulated blood at the same time. 

Statistical analyses were performed using Statistica 11.0 (StatSoft Inc., Tulsa, OK, USA) or Statistical Package for the Social Sciences v26 (I.B.M., Armonk, NY, USA). Visualization was carried out using Office 365 (Microsoft, Seattle, WA, USA) or Pizma (GraphPad Software (Version is 10.1.1), Boston, MA, USA).

## 3. Results

### 3.1. Patient Characteristics

Demographic, clinical, and outcome data are summarized in Table 1. Our sample was predominantly male (77%) and white (87%), with diabetes being the most common comorbidity (29%). Fifty-seven patients underwent CABG; 87% of all study patients were discharged home after an average hospital stay of 10 days.

### 3.2. Responsiveness of the Whole Blood to Bacterial and Viral Pathogens Evolves after Cardiac Surgery

Production of IL-2 by the whole blood in response to applied stimuli was generally low and highly variable (Figure 1A). In the univariate mixed model, only time, but not stimulation, had a significant effect on IL-2 production, with a tendency to increase at seven days and three months across all the stimulations in the two-way ANOVA model (F[4] = 3.62; *p* = 0.007). The unstimulated serum had a borderline tendency to contain more IL-2 at three months than the pre-op serum (Figure 1B). H3N2 stimulation resulted in a significant increase (W[16] = 70; *p* = 0.047) at t_3m_ only (Figure 1C). FluBlok and LPS tended to stimulate the IL-2 production at three months and seven days, but the effects were highly variable (Figure 1D,E). Regression analysis demonstrated that unstimulated levels of IL-2 at three months contribute to the variance of IL-2 in the whole blood stimulated by LPS (*p* = 0.0046) but not by H3N2 (*p* = 0.065) or FluBlok (*p* = 0.51). FluBlok triggered a statistically significant response only in blood collected seven days after the surgery compared to an unstimulated sample obtained at the same time (data not shown).

IFNγ production by the whole blood was significantly elevated after LPS exposure, particularly at 7 days and 90 days (Figure 2). In contrast, stimulation with H3N2 and FluBlok did not elicit a response above the background (Figure 2).

IL-6 level in whole blood was dependent on the stimulus (F[3] = 24.019; *p* < 0.001; *η*^2^ = 0.66), time (F[3] = 17.72; *p* < 0.001; *η*^2^ = 0.5) and interaction between these two elements (F[9] = 11.985; *p* < 0.001; *η*^2^ = 0.31), as demonstrated in the univariate mixed model (Figure 3A). Unstimulated blood demonstrated an increase in IL-6 at t_24h_ and t_7d_. However, H3N2, FluBlok, and LPS increased IL-6 production at t_7d_ (Figure 3A–D). Interestingly, the response to LPS was bimodal at almost any time point, with a tendency to merge into the baseline (Appendix A).

Production of TNFα was dependent on time (F[3] = 5.1; *p* = 0.02; *η*^2^ = 0.18) and stimulation (F[3] = 11.817; *p* < 0.001; *η*^2^ = 0.39). In addition, there was an interactive dependence between these two variables (F[9] = 30.086; *p* < 0.0001; *η*^2^ = 0.31) in the univariate model (Figure 4A). Unstimulated blood tended to have elevated TNFα levels at seven days and three months (Figure 4A). TNFα production in response to H3N2 was significantly increased at three months (Figure 4A). Flublock tended to increase TNFα production at seven days, while the increase was borderline at three months (Figure 4A). Contrary to our expectations, TNFα response to LPS stimulation appeared to be suppressed at t_24h_ (W[44] = −243; *p* = 0.034). IL-10 production by the whole blood was not significantly induced in our experiments (Figure 4B). Production of IL-10 was dependent on time (F[3] = 4.58; *p* = 0.04; *η*^2^ = 0.46) and stimulation (F[3] = 15.284; *p* < 0.001; *η*^2^ = 0.14) with a weak interactive dependence between the two (F[9] = 5.41; *p* < 0.001; *η*^2^ = 0.145) strength of (Figure 4B). M-CSF production by the whole blood was dependent only on time (F[3] = 21.419; *p* < 0.001; *η*^2^ = 0.12) but not on stimulation, with no significant interactive dependence between these two variables (Figure 4C). The most significant variability of M-CSF production in response to various stimuli was at t_7d_, with spontaneous production significantly enhanced even in unstimulated cells (Figure 4C). The unstimulated t_7d_ production of M-CSF correlated with H3N2 (*r*^2^ = 0.88; *p* = 0.001), FB (*r*^2^ = 0.71; *p* = 0.001), and LPS (*r*^2^ = 0.88; *p* = 0.001).

### 3.3. Other Cytokine Levels in Studied Whole Blood Serum

We found no statistically significant differences in IL-1β, IL-5, IL-7, IL-8, IL-12p70, IL-13, IL-31, RANTES, TNFβ, and IL-18 levels after stimulations. However, MIP-1a, IL-27, IL-4, MCP-1, eotaxin, IL-17a, and IL-9 demonstrated significant variability over stimulus and time (Figure 5).

### 3.4. Dynamic Changes in Markers of the Inflammation in the Peri-Surgical Period

White cell count demonstrated an increase in the absolute count with normalization at three months (F[4;339] = 87; *p* = 0.00001) (Figure 6A). With regard to platelet count, thrombocytopenia was observed at 24 h and 7 days, followed by thrombocytosis at 3 months (F[3;195] = 87; *p* = 0.0001) (Figure 6B).

IL-6 and IL-8 increased at 24 h and 7 days but returned to baseline at three months (Figure 6C,D). CRP serum level increased at t_24h_ and t_7d_ but remained elevated at t_3m_ (F[3;25] = 32.4; *p* = 0.00001) compared to the presurgical baseline (Figure 6D).

### 3.5. Perioperative Management and Cytokine Production

Compared to other cardiac surgeries, coronary artery bypass graft (CABG) was associated with an increased production of IL-6 at 24 h (IL-6_CABG_ = 609.7 ± 932.8 vs. IL-6_NoNCABG_ = 181.9 ± 232.4; U = 378; *p* = 0.009) in response to H3N2 while IL-2 production decreased at 3 months (IL-2_CABG_ = 5.67 ± 10.48 vs. IL-2_NoNCABG_ = 19.47 ± 43.02; U = 66.50; *p* = 0.049).

Duration of anesthesia and surgery, estimated blood loss, fluids, blood product, and resuscitation did not correlate with the production of measured cytokines after any stimulus.

Perioperative application of acetaminophen, ketorolac, or acetylsalicylic acid did not affect IL-2, IL-6, IL-10, TNFα, or M-CSF production at 24 h or 7 days.

### 3.6. Whole Blood Simulation and Clinical Outcome

IL-2 production at t_24h_ stimulated by FluBlok correlated with APACHE_1h_ (Figure 7). IL-2 production after H3N2 and LPS correlated with APACHE_24h_ (Figure 7). APACHE_24h_ and APACHE_48h_ correlated significantly with the production of IL-10 induced by H3N2, FluBlok, and LPS APACHE at 24 and 48 but not with APACHE_1h_ (Figure 7).

Length of stay in the ICU did not correlate with the production of measured cytokines at 24 h and 7 days in response to any stimuli. Disposition (home vs. facility) was not affected by cytokine production at any time. Low incidence of death (*n* = 1), pulmonary embolism (*n* = 5), and deep venous thrombosis (*n* = 5) in the studied cohort precluded meaningful statistical analyses of those data.

## 4. Discussion

This is the first study to utilize whole blood stimulation of adult patients undergoing cardiac surgery to describe its immunological reactivity to viral and bacterial pathogens with three-month follow-ups. We specifically focused on long-term follow-up to detect recovery of whole blood responsiveness to baseline, since most other studies reported a much shorter observation period [2,13,25,39,40,41,42,43,44]. We hypothesized that the ability of the whole blood to produce cytokines would be elevated shortly after cardiac surgery with the emergence of global suppression at three months [22,26,35,56,57]. Additionally, the resolution of the elevated cytokine production would accompany the return of non-specific inflammatory markers to their baseline level. The results were much more complex. We found that LPS had the most impact on TNFα, IL-2, and IL-10 production at various time points after surgery. A decline in production of INFγ was seen, while M-CSF was uniformly elevated at 7 days irrespective of the stimulus.

Contrary to our expectation, hyperresponsiveness was not seen late during convalescence [8,18,22,26,34,36]. M-CSF and IL-6 production was elevated at seven days in response to all the stimuli, suggesting a compensatory response. Uniform release of M-CSF at 7 days may signify the emergence of immuno-suppressive or alternatively activated cells, but the timing of that phenomenon occurred earlier than previously suggested [17]. Even more remarkably, response to the bacterial and viral stimulation varied over time; cytokine production did not return to the presurgical baseline. Initial clinical acuity appeared to have the strongest correlation with IL-2 and IL-10 production in response to viral pathogens. Similar findings with respect to IL-10 production were previously reported [10,25,29,35,50]. Testing of a patient’s immune reactivity provided additional data while using simple and feasible methodology.

We examined the correlation between the production of cytokines based on the whole blood and measurements of initial clinical acuity. However, no clear correlates were found, in contrast to the prior studies showing a correlation with T cell- or monocyte-related cytokines in serum [5,7,10,24,45]. A positive relationship between IL-10 and APACHE II score during the first 48 h may be a demonstration of bypass-induced IL-10 secretion and subsequent LPS-induced immunosuppression with potential unfavorable clinical outcomes [5,8,10,22,29,46,56]. While IL-10 production may represent diminished tolerance to an exaggerated immunological response, it can also result in immunosuppression. This effect can be further augmented by M-CSF, IL-4, and IL-27 [24,61,62,63,64]. Clinically, this may manifest in increased susceptibility to perioperative infection or the emergence of opportunistic infections, including CMV.

These changes in whole blood cytokine production need to be contrasted with traditional markers of immune system activation. We noticed leukocytosis, thrombocytosis, and elevated serum CRP at three months after the surgery, with variable production of different cytokines in response to the stimuli. This suggests that whole blood stimulation may represent a very different assay for immune system performance. For example, despite elevated levels of serum IL-6 in the perioperative period, IL-6 production by the whole blood following stimulation was not significantly altered [43]. This may suggest a different source of IL-6 in the blood than leukocytes including endothelial or adipose cells [1,2,26].

This was a pilot study, and its findings are intended to be hypothesis-generating. There are some methodological biases to be accounted for. We could not quantify the effect of variable doses of heparin, dexamethesone, and furosemide during the preparation for the initiation of cardiopulmonary bypass or during the blood collection [65,66,67,68]. Some of these agents can stimulate monocyte and other cell populations. We did not report such data as hyperglycemia or the level of vitamin D despite their significant influence on immune performance [63]. As the leukocyte count and composition vary after a surgery [9,10,45,46,48,69], production of cytokines can be affected by the total number of cells available to respond to a given stimulation. However, cytokine production was dependent on the timing and the given stimulus, suggesting that the total number of leukocytes was not a dominant factor in their production. We did not characterize the leukocyte profile at each study time point. We used a predetermined amount of LPS, which was intended to approximate the dose present during a bacterial challenge [44]. H3N2 load during viremia was not characterized in vitro. Finally, using FluBlok exposed blood to several boosters and additives that are not typically found during influenza viremia [59,60]. We aimed to create a testing system that closely resembled the clinical condition occurring during the perioperative period when a patient can be rechallenged by bacterial and viral pathogens, leading to secondary perioperative infections. However, the exact dose and duration of the patient’s immune system exposure to these pathogens is unclear. Whole blood stimulation also cannot account for cytokine production by the endothelium and other immunologically active cells [19]. A lack of a predictive value may reflect the redundancy of the whole blood [38,51].

The study has several strengths. Our assay factors in neutrophils and platelets that contribute to overall cytokine production in the whole blood. We controlled for key critical variables such as perioperative management and postoperative care. Our data on a diverse population of adult individuals adds to the existing literature describing immunological responses described mostly in pediatric surgery patients [1,2,8,37,39,40,57]. The observation window extended well beyond the traditional 28 days reported in most prior studies [2,8,10,37,39,42,57]. The sample size was relatively large for an observational pilot study. Utilized reagents were standardized, and the techniques used to measure cytokines were robust. Changes in cytokine production detected an emergence of hyporesponsiveness, a phenomenon with potential clinical significance [13,21,25,36]. Also, prior observations of IL-2 and IFNγ production and their overall performance after cardiac surgery suggest a potential link to relevant clinical outcomes [10,50].

The next step is standardizing the whole blood stimulation testing and addressing the variability in cytokine responses. Conducting in-cellular cytokine production staining may allow for identification of the primary producers of different cytokines. Designing a study to look for delayed clinical correlates is also warranted. This may be the most significant potential of the assays, as several cytokines are linked to the emergence of long-term complications.

## 5. Conclusions

Utilization of the whole blood to measure cytokine production as immune system responsiveness provides a unique insight into the functioning of the immune system following cardiac surgery.

## Figures and Tables

**Figure 1 biomedicines-12-00028-f001:**
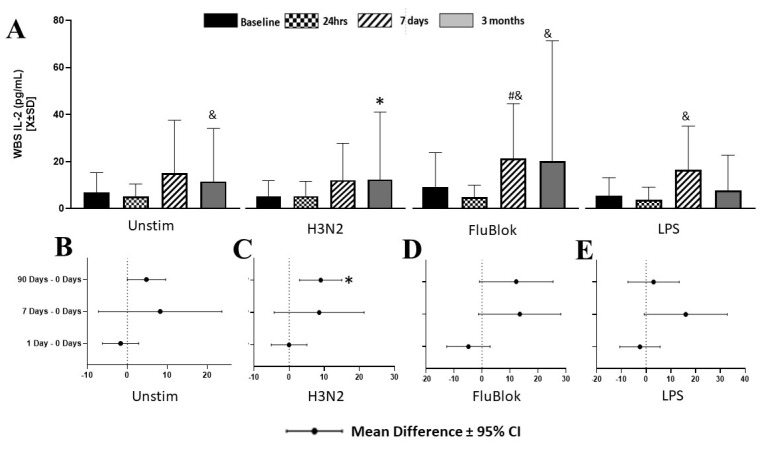
Production of IL-2 by whole blood in response to various infectious pathogens demonstrated significant variability across time and stimuli, with several changes having borderline significance (**A**). More specifically, unstimulated blood (**B**) demonstrated a marginal increase in IL-2 production at three months compared to unstimulated samples. H3N2 significantly increased IL-2 production at three months compared to presurgical samples, though this increase was not higher than the increases observed in unstimulated samples (**A**,**C**). FluBlok (**A**,**D**) and LPS (**A**,**E**) had borderline effects at the post-stimulation concentration of IL-2 at seven days and three months. * denotes the significance level below 0.05 utilizing a two-tailed hypothesis when comparing the same stimuli over time to a presurgical baseline. & denotes a significance level below 0.05, utilizing a one-sided specific hypothesis when comparing the same stimuli over time. # denotes the significance level below 0.05 in two-tailed testing while comparing cytokine production to unstimulated blood at the same time.

**Figure 2 biomedicines-12-00028-f002:**
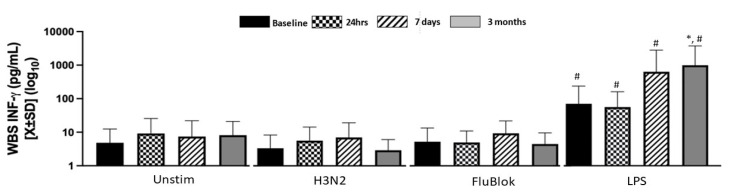
IFNγ production by WBS only from LPS exposure resulted in a significant increase across all time points, particularly at 7 days and 3 months compared to the presurgical level and the level when unstimulated. H3N2 and FluBlok did not elicit a significant response. * denotes a significance level below 0.05, utilizing a two-tailed hypothesis when comparing the same stimuli over time to a presurgical baseline. # denotes a significance level below 0.05 in two-tailed testing while comparing cytokine production to unstimulated blood at the same time.

**Figure 3 biomedicines-12-00028-f003:**
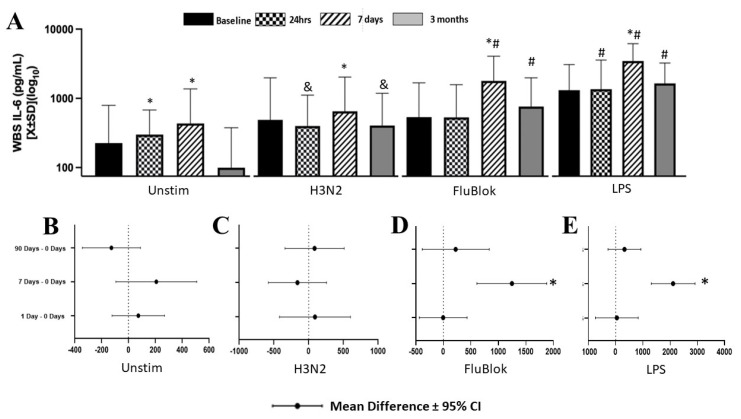
IL-6 cytokine level demonstrated significant changes depending on stimulation and post-surgical time point (**A**). IL-6 in unstimulated samples was increased at 24 h and 7 days but not at 3 months (**A**,**B**). H3N2 (**A**,**C**), Flublock (**A**,**D**), and LPS (**A**,**E**) showed an increase at seven days compared to unstimulated samples at the same time in the perisurgical period. * denotes a significance level below 0.05, utilizing a two-tailed hypothesis when comparing the same stimuli over time to a presurgical baseline. & denotes a significance level below 0.05, utilizing a one-sided specific hypothesis when comparing the same stimuli over time. # denotes a significance level below 0.05 in two-tailed testing while comparing cytokine production to unstimulated blood at the same time.

**Figure 4 biomedicines-12-00028-f004:**
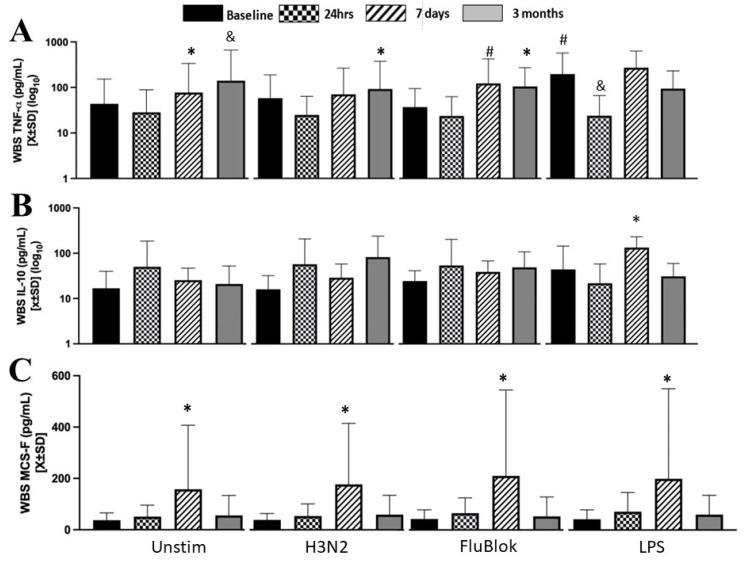
Stimulation of the whole blood with viral and bacterial pathogens revealed three distinctive patterns. TNFα production was strongly induced at seven days and three months compared to baseline (**A**). Induction of IL-10 was limited to LPS at t_7d_ time point (**B**). Finally, the production of M-CSF by the whole blood was signified by increased production at t_7d_ in unstimulated samples or after exposure to any viral or bacterial pathogens (**C**). * denotes a significance level below 0.05, utilizing a two-tailed hypothesis when comparing the same stimuli over time to a presurgical baseline. & denotes a significance level below 0.05, utilizing a one-sided specific hypothesis when comparing the same stimuli over time. # denotes a significance level below 0.05 in two-tailed testing while comparing cytokine production to unstimulated blood simultaneously.

**Figure 5 biomedicines-12-00028-f005:**
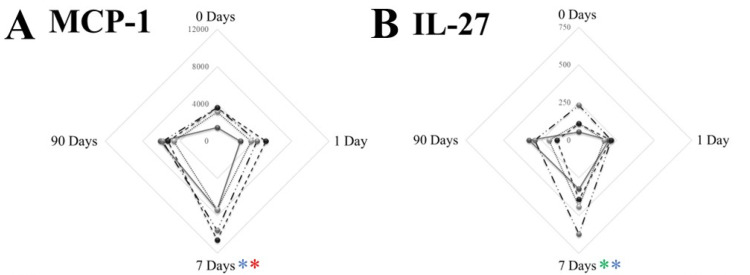
Stimulation of the whole blood with viral and bacterial pathogens. MCP-1 (**A**), IL-27 (**B**), IL-4 (**C**), and MIP-1α (**D**) showed different production levels, which seem primarily dependent on the stimulus. (**A**) MCP-1 was significantly different from baseline at 7 days with FB and LPS stimulation. (**B**) IL-27 was significantly different from baseline at 7 days in the H3N2 and FB groups. (**C**) IL-4 only differed from baseline at 7 days in the H3N2 group. (**D**) LPS stimulation revealed significant changes in MIP-1α at 7 days and 90 days compared to baseline. * denotes the level of significance below 0.05, utilizing a two-tailed hypothesis when comparing the same stimuli over time. Different colors differentiate between different stimuli (H3N2—green asterisk, FB—blue asterisk, and LPs—red asterisk; unstimulated samples showed no differences and were not color-coded).

**Figure 6 biomedicines-12-00028-f006:**
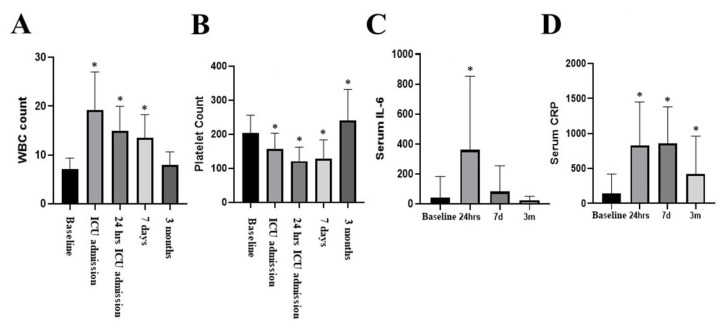
White cell count (**A**). Platelet count (**B**). IL-6 and IL-8 increased at 24 h and returned to baseline at 3 months (**C**,**D**). CRP serum level increased initially and remained elevated at 3 months (**D**). * denotes a level of significance below 0.05, utilizing a two-tailed hypothesis when comparing the same variable over time to the baseline.

**Figure 7 biomedicines-12-00028-f007:**
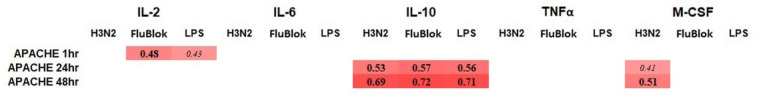
Correlation between APACHE score and selected serum cytokines level. Fields in red denote statistically significant *r*^2^.

**Table 1 biomedicines-12-00028-t001:** Patient sample demographics, clinical data, and outcomes.

**Demographics (*N* = 90)**		
Age (Mean ± SD) [Years]		64.1 ± 12.2
	Over 60 Years [%]	72.22%
Sex	Male [%]	76.85%
	Female [%]	23.15%
Race	African American [%]	3.70%
	White [%]	87.04%
	Other/Asian/Unknown [%]	9.26%
**Pre-Existing Conditions**		
Weight (Mean ± SD) [kg]		38.76 ± 9.30
BMI (Mean ± SD)		28.34 ± 5.92
Charlson Comorbidity Index (Mean ± SD)		3.80 ± 2.08
ACS/MI [%]		10.20%
CHF [%]		19.40%
PVD [%]		9.30%
CVA/TIA [%]		10.20%
COPD [%]		7.40%
DM [%]		28.70%
**Anesthesia & Surgery Data**		
Duration of anesthesia (Mean ± SD) [min]		371.5 ± 100.7
Duration of surgery (Mean ± SD) [min]		264.8 ± 92.84
Duration of cardiopulmonary bypass (Mean ± SD) [min]		128.2 ± 60.71
Coronary artery bypass surgery [*n*]		57
Mitral valvuloplasty & replacement [*n*]		81
Aortic valvuloplasty & replacement [*n*]		75
Aortic aneurysm repair [*n*]		12
Estimated Blood Loss (Mean ± SD) [mL]		198.5 ± 283.6
**Perioperative management**		
*Transfusions during surgery*		
Packed Red Blood Cells (Mean, IQR) [mL]		99.06, 244
Fresh Frozen Plasma (Mean, IQR) [mL]		51.98, 194.3
Total crystalloid during surgery (Mean, IQR) [mL]		1332, 619.9
*Clinical Care during 24 h post-surgery*		
Packed Red Blood Cells (Mean, IQR) [mL]		22.86, 99.28
Fresh Frozen Plasma (Mean, IQR) [mL]		8.57, 87.83
Corticosteroid Administration [% of all cases]		12.96%
Ketorolac Administration [% of all cases]		8.33%
Acetaminophen Administration [% of all cases]		79.62%
Acetylsalicylic acid Administration [% of all cases]		75.00%
Opioids Administration (Mean ± SD) [mg]		704 ± 219
Benzodiazepine administration (Mean ± SD) [mg]		3.78 ± 1.95
*Outcome at 28 days*		
LOS ICU (Mean ± SD) [Days]		7.64 ± 40.89
LOS Hospital (Mean ± SD) [Days]		9.55 ± 20.79
Discharged home/In the healthcare facility/expired [%]		87.3%/6.39%/1.26%

## Data Availability

Access to the data will be granted upon reasonable request after IRB approval.

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
