# Peer review of "Whole Blood Reactivity to Viral and Bacterial Pathogens after Non-Emergent Cardiac Surgery during the Acute and Convalescence Periods Demonstrates a Distinctive Profile of Cytokines Production Compared to the Preoperative Baseline in Cohort of 108 Patients, Suggesting Immunological Reprogramming during the 28 Days Traditionally Recognized as the Post-Surgical Recovery Period"

_biomedicines, 2023, doi:10.3390/biomedicines12010028_

Round 1

Reviewer 1 Report

Comments and Suggestions for Authors

Laudanski et al in present manuscript studied the whole blood reactivity to viral as well bacterial pathogens after cardiac surgery during a 3-month period post surgery. Reactivity was tested by measuring various inflammatory as well as anti-inflammatory cytokines in blood serum after in vitro stimulation along with white blood and platelets counts. 

The study has many pitfalls regarding the presentation of results and their statistical analysis. Specifically, 

1. The authors should present the method(s) followed for the detection is missing in Materials and Methods section.

2. In the results section, the authors did not make comparisons between unstimulated and stimulated samples in the respective time points which is significant since it seems that stimulation with either stimulus applied did not trigger in the most cases the production of cytokines. This is evidenced by the cytokine levels detected in the stimulated samples which were not higher than the unstimulated samples. Thus, these comparisons should be applied for all cytokines tested. Also, in order to make correlations between time and cytokine levels in each stimulation, a regression analysis should be applied and presented, respectively.

3. The figure legends are insufficient. They should stand alone describing the method followed, the samples and the statistical analysis applied. Thus, all the legends should be written again in a more concise way. Why authors take as significant the level below 0.5 since in every test applied the significance is taken in levels below 0.05? This is important because in this case, it seems that stimulation in the most cases did not trigger the induction of cytokine production at levels over baseline levels.

Specifically, in figure 1, information describing panel (A) is not correct. Also, details describing panels B, C, D and E are missing. In figure 3 explanation of panels B, C and D is missing. Figure 4 should be more descriptive. 

4. Results presented in paragraph 3.3 need to be described regarding time and stimulus and the respective figure 5 should more detailed. 

5. Paragraph 3.4: Why authors supported that CRP levels decreased at 7 days, since in the respective figure 6E the levels are similar to those detected at 24 h post surgery?

6. Figure 6: Results presented in panel F are missing in the respective Results section. Moreover, the legend should more descriptive. 

7. Paragraph 3.5: Results be presented in a correlogram.

Author Response

1. The authors should present the method(s) followed for the detection is missing in Materials and Methods section.

We incorporated the required details. We appreciate this request from the reviewer.

2. In the results section, the authors did not make comparisons between unstimulated and stimulated samples in the respective time points which is significant since it seems that stimulation with either stimulus applied did not trigger in the most cases the production of cytokines. This is evidenced by the cytokine levels detected in the stimulated samples which were not higher than the unstimulated samples. Thus, these comparisons should be applied for all cytokines tested. Also, in order to make correlations between time and cytokine levels in each stimulation, a regression analysis should be applied and presented, respectively. 

We conducted analysis to look at contrast to stimulated vs. unstimulated samples. These analyses are marked with # on the figures. Lack of response to some of the stimuli should not be surprising. We used a much lower concentration of biological agents. However, these stimuli may be effective if the leukocytes are primed as those observed shortly after surgery. Using lower concentration of the stimuli results in more physiological response while avoiding necrosis and apoptosis induced by hyper-concentrated stimulus.

3. The figure legends are insufficient. They should stand alone describing the method followed, the samples and the statistical analysis applied. Thus, all the legends should be written again in a more concise way. Why authors take as significant the level below 0.5 since in every test applied the significance is taken in levels below 0.05? This is important because in this case, it seems that stimulation in the most cases did not trigger the induction of cytokine production at levels over baseline levels.  

We improved figure legends along with several of them being redrawn or provided in higher resolution. Standarized document for all figure legends were provided.

4. Results presented in paragraph 3.3 need to be described regarding time and stimulus and the respective figure 5 should more detailed.

We narrowed down the number of displayed subfigures to the one with significance. Figures were enlarged and simplified. Additional details were added. The description was improved. We also added color to better reflect the stimulation types visualized in the figures.

5. Paragraph 3.4: Why authors supported that CRP levels decreased at 7 days, since in the respective figure 6E the levels are similar to those detected at 24 h post surgery? 

We corrected our mishandling of this information. In addition, graphs have been reformatted to align with the format used in other figures. We also removed IL-8 as being redundant.

6. Figure 6: Results presented in panel F are missing in the respective Results section. Moreover, the legend should more descriptive. 

This graph has been upgraded and rephrased.

7. Paragraph 3.5: Results be presented in a correlogram. 

These comparisons are an analysis of the difference between two different conditions. We believe that the current data presentation is more accurate.

Reviewer 2 Report

Comments and Suggestions for Authors

The research study titled, "Distinct Cytokine Profiles in Whole Blood Reactivity to Pathogens Post Non-Emergent Cardiac Surgery: Acute and Convalescence Phases," aimed to explore the shifts in immune reactivity following cardiac surgery by analyzing whole blood. It is postulated that the capacity of whole blood to produce cytokines would spike soon after the surgery. This surge would be marked by a notable suppression at the seventh day and subsequently return to its presurgical levels. It is opted to measure both proinflammatory (IL-6, TNFα, IL-2, IFNγ) and anti-inflammatory cytokines (IL-10, M-CSF) given their pivotal role in restoring immune balance.

The patterns of cytokine production were compared with conventional immune system activation markers, including white blood count (WBC), platelet count, serum C-reactive protein, and serum levels of interleukin-6 (IL-6) and interleukin-8 (IL-8).

While the paper is comprehensively written, it prompts a few questions:

How do these findings translate clinically?

How does the study compare with the current state-of-the-art?

What novel insights or contributions does this research offer?

Author Response

1. How do these findings translate clinically? 

We rephrased the discussion to reflect clinical relevance and included the necessary references.

2. How does the study compare with the current state-of-the-art? 

We introduce a comparison to other studies, yet only a few of them were done and none on the population of cardiopulmonary bypass patients.

3. What novel insights or contributions does this research offer? 

The discussion and introduction were changed and re-written to stipulate the unique findings of the study better.

Round 2

Reviewer 2 Report

Comments and Suggestions for Authors

Authors have performed appropriate modifications.